# The Role of Polyphenols in Modulating PON1 Activity Regarding Endothelial Dysfunction and Atherosclerosis

**DOI:** 10.3390/ijms25052962

**Published:** 2024-03-04

**Authors:** Teodora Bianca Sirca, Mariana Eugenia Mureșan, Annamaria Pallag, Eleonora Marian, Tunde Jurca, Laura Grațiela Vicaș, Ioana Paula Tunduc, Felicia Manole, Liana Ștefan

**Affiliations:** 1Doctoral School of Biomedical Sciences, University of Oradea, No. 1 University Street, 410087 Oradea, Romania; sircateodora@yahoo.com (T.B.S.); mmuresan@uoradea.ro (M.E.M.); 2Department of Pharmacy, Faculty of Medicine and Pharmacy, University of Oradea, 29 Nicolae Jiga Street, 410028 Oradea, Romania; apallag@uoradea.ro (A.P.); emarian@uoradea.ro (E.M.); tjurca@uoradea.ro (T.J.); 3Department of Cardiology, Clinical County Emergency Hospital of Bihor, Gheorghe Doja Street 65-67, 410169 Oradea, Romania; paublaj@yahoo.com; 4Department of Surgery, Faculty of Medicine and Pharmacy, University of Oradea, 1st December Square 10, 410073 Oradea, Romania; fmanole@uoradea.ro (F.M.); lianaantal@gmail.com (L.Ș.)

**Keywords:** paraoxonase 1 (PON1), endothelial dysfunction, oxidative stress, dyslipidemia, atherosclerosis, polyphenols, cardiovascular disease

## Abstract

The incidence and prevalence of cardiovascular diseases are still rising. The principal mechanism that drives them is atherosclerosis, an affection given by dyslipidemia and a pro-inflammatory state. Paraoxonase enzymes have a protective role due to their ability to contribute to antioxidant and anti-inflammatory pathways, especially paraoxonase 1 (PON1). PON1 binds with HDL (high-density lipoprotein), and high serum levels lead to a protective state against dyslipidemia, cardiovascular diseases, diabetes, stroke, nonalcoholic fatty liver disease, and many others. Modulating PON1 expression might be a treatment objective with significant results in limiting the prevalence of atherosclerosis. Lifestyle including diet and exercise can raise its levels, and some beneficial plants have been found to influence PON1 levels; therefore, more studies on herbal components are needed. Our purpose is to highlight the principal roles of Praoxonase 1, its implications in dyslipidemia, cardiovascular diseases, stroke, and other diseases, and to emphasize plants that can modulate PON1 expression, targeting the potential of some flavonoids that could be introduced as supplements in our diet and to validate the hypothesis that flavonoids have any effects regarding PON1 function.

## 1. Introduction

Cardiovascular diseases have a high prevalence worldwide and are one of the main causes of mortality. It is well known that many factors are responsible for these diseases, and the focus should also be on preventing them; therefore, we must understand the molecular pathways. The process behind myocardial infarction, ischemic cardiomyopathy, and peripheral arterial disease is called atherosclerosis. The vascular endothelium involved in heart disease, peripheral vascular disease, diabetes, and venous thrombosis comprises one layer of specialized cells [1]. It contains multiple receptors for different molecules such as proteins, lipids, hormones, and metabolites. Endothelial cells are responsible for normal blood flow, controlling thrombosis and thrombolysis, platelet aggregation/activation, and vascular tone [1]. An anticoagulant and antiplatelet mechanism maintains blood fluidity. A breach in any mechanism can lead to thrombosis or bleeding [1]. Endothelial cells are implicated in an acute inflammatory response known as the cardinal signs of inflammation or type I activation calor, rubor, dolor, and tumor, a self-restricted response with no important changes. Type 2 activation means a bigger response to certain pathogens that use some pathways such as NF-kB, leading to changes in function and morphology [2].

In the last decade, significant progress has been made in establishing the pharmacological mechanisms of medicinal plants and the individual chemical constituents responsible for them. Extracts from various parts of the plants show therapeutic properties, and several studies have reported medicinal benefits based on biochemical mechanisms.

Various studies in the literature have shown that a series of bioactive compounds from plants such as polyphenolic derivatives (procyanidins, quercetin, and flavonols) have rapid absorption in the plasma, having maximum plasma concentrations 2–3 h after ingestion. Their biological activity has been proven through various properties: antioxidant, antiallergic, anti-inflammatory, antiviral and antimicrobial, antiproliferative, antimutagenic, and anticarcinogenic, blocking free radicals, regulating cell cycle arrest, apoptosis, and the induction of antioxidant enzyme activity; more interestingly, polyphenols can modulate several important cell signaling pathways, such as nuclear factor kappa-B (NF-kB), transcription factor AP-1 (activator protein-1), extracellular signal-regulated protein kinase (ERK), phosphoinositide 3 kinase B/protein kinase (Akt), mitogen-activated protein kinases (MAPKs), and nuclear factor erythroid-related factor 2 (Nrf2).

Some studies suggest that in various chronic diseases, the activity of paraoxonase 1 (PON1) is decreased, being found simultaneously with the decrease in superoxide dismutase (SOD) and increase in catalase (CAT). A positive correlation was also observed between CAT/SOD, CAT/PON1 ratios, and various parameters specific to each pathology. Thus, it is suggested by the researchers to measure the biomarkers of the cellular redox status [3,4].

## 2. Materials and Methods

Our concerns were based on a review of experimental research using bibliometric databases as sources of scientific documentation to identify new scientific trends through a literature study report on the activity of plant extracts, specifically PON1 activity. Thus, the databases relevant to this purpose were taken into consideration: Web of Science—Core Collection and Scopus. In Figure 1 we have presented a distribution for the items included in our documentation.

To identify scientific articles published on paraoxonase 1 in cardiovascular disease, the articles were searched using representative keywords to locate the primary data, outcomes, and papers in the field. To search for articles, the following keywords were used: *paraoxonase 1*; *cardiovascular disease*. A total of 2307 scientific articles were identified. A Prisma flow diagram was used for a description of how to select the studies and articles included in the review is shown in Figure 1 [5,6].

Recent findings show the role of plant extracts through the ability of various bioactive compounds in their compositions to stimulate the activation of PON1 transcription.

## 3. Paraoxonase 1 and Its Implications in Various Diseases

### 3.1. Endothelial Dysfunction

Endothelial dysfunction describes a state when the normal proprieties of the endothelium are disturbed, leading to a pro-inflammatory and prothrombotic state, abnormal vasodilatation, leukocyte adhesion, and higher permeability [1,7]. Nitric oxide is a soluble gas with a major role in vascular homeostasis, being a strong vasodilatory, anti-inflammatory, and antioxidant substance [8]. Free radicals can damage the balance of NO concentration and the endothelial barrier, allowing other substances to enter the body [1]. Increased levels of reactive oxygen species (ROS) are responsible for cell damage and death resulting from an imbalance in pro- and antioxidative species in favor of pro-oxidant ones. Oxidative stress is behind many diseases due to its major role in inflammation and fibrogenesis [9]. Endothelial dysfunction drives peripheral vascular disease through abnormal NO levels, which means an increased state of vasoconstriction, leading to ischemia and therefore a discordant oxygen delivery demand. Plaque rupture also results from endothelial dysfunction, generating critical limb ischemia [1]. NO levels are maintained in normal parameters, but when dysfunction occurs, lower or higher levels lead to different outcomes; for example, during myocardial infarction, high NO levels have negative inotropic effects [8].

Platelets are a key member of atherosclerosis, being the main cells in the first stage of thrombus formation. The adhesion, activation, and aggregation of platelets do not occur if the endothelium is intact, only when it has a breach or in other specific situations. When the endothelium is exposed, it interacts with the endothelium, collagen, thromboxane, and ADP (adenosine diphosphate), and thrombin activates them. As a result, they secrete chemotaxins, coagulation factors, and vasoconstrictors, inducing more platelet activation [1]. Although for a long time it was believed that platelets only adhere to the endothelium if there is damage, the reality is that they can stick even if it is inflamed or if it is a location that is prone to adhesion such as the carotid bifurcation [1].

Hypertension is also combined with endothelial dysfunction, but it remains unknown if it is the cause or result. At first, it was believed that endothelial dysfunction occurs because of the chronic increased pressure, but studies on hypertensive drugs showed no changes in endothelial dysfunction [1]. A newer study suggests that the effect depends on the antihypertensive classes, because using angiotensin receptor blockers or angiotensin-converting enzyme inhibitors does fight against the dysfunction [10]. High arterial pressure leads to an overproduction or a more ample effect of vasoconstriction agents in contrast to vasodilative factors [1]. NO (nitric oxide) metabolism is also dysbalanced through oxidative stress mediated by angiotensin II. Angiotensin II is responsible for the disturbance between oxidants/antioxidants by increasing the permeability of the vascular lumen [7,11].

An association between endothelial dysfunction leading to inflammation and atherosclerosis is given by atherosclerosis-associated endothelial–leukocyte adhesion molecule (VCAM-1). In animal models, it has been found that in coronary arteries, VCAM-1 lays on the intact endothelium, surfacing the plaque, and is usually found in areas prone to lesions. Moreover, VCAM-1’s isoform plasma levels correlate with more serious atherosclerotic lesions [2].

All the factors implicated in atherosclerosis (smoking, hypertension, diabetes, and hypercholesterolemia) are combined with endothelial dysfunction resulting in a pro-inflammatory state, probably due to lower NO levels and therefore increased leukocyte adhesion molecules and cytokines leading to a state prone to arterial lesions.

Firstly, macrophages are internalized into the intima and transformed into lipoprotein and foam cells. Chemokines induce the activity of muscle cells generating a fibromuscular plaque, which will eventually become a fibrous cap with oxidized lipoproteins and inflammatory cells modulating the endothelium and finally leading to an unstable plaque [2]. As Ulf Landmesser and David G. Harrison suggested, lower NO arterial availability is caused by superoxide production, which either degrades NO or the cofactor used in its synthesis [1,12]. Lipids also diminish NO levels because of their power to activate pro-inflammatory pathways like those mediated by NF-kB (nuclear factor kappa-B) [13]. Endothelin plays a role in atherogenesis by the stimulation of pro-inflammatory mediators (IL-6, IL-8, TNF-α (tumor necrosis factor), and superoxide anion), smooth muscle hypertrophy, and increased response to angiotensin II [1,14]. TNF-α stimulates ROS formation, and in coronary arteries, this implies the transformation of stable plaque into unstable plaque, which can lead to a coronary event (Figure 2) [15].

Atherosclerotic lesions can transform into a vulnerable plaque that can break down and lead to thrombosis. Depending on the dimensions of the thrombus and of the vessel, it can lead to stenosis or occlusion, and ultimately end-organ damage [16]. A question that remains in our mind is how we can assess endothelial dysfunction. The principle is to measure the ability of the cell to stimulate vasodilation or vasoconstriction. This was studied by Sandoo et al. using pulse wave velocity, venous occlusion plethysmography, and flow-mediated vasodilatation in the brachial artery [17,18]. The last one is based on NO (nitric oxide) release, by measuring the diameter of the brachial artery, inducing ischemia with a cuff, and after 5 min, NO causes vasodilatation. This result can be seen when we re-measure [15]. Also, the carotid–intima media thickness is used more frequently because it is more accessible, easier to measure, and non-invasive [18]. Newer methods such as epicardial fat thickness, the ankle–brachial index, and arterial stiffness were used for the assessment of endothelial dysfunction [18]. Another way is to measure biomarkers such as IL-1β, IL-6, E-selectin, or VCAM-1-associated CRP (C reactive protein), which are increased in atherosclerotic lesions that can be seen using paraclinical examination such as Doppler echography for peripheral lesions or coronagraphy for coronary atherosclerosis. Other tests used are ischemia-modified albumin (IMA), endothelial cell-specific molecule 1 (Endocan), pentraxin-3 (PTX3), asymmetrical dimethylarginine (ADMA), endothelial microparticles, endothelial progenitor cells (EPCs), angiopoietin-1 (Ang-1), and von Willebrand Factor (vWf) [18]. However, the standard remains FMD (flow-mediated vasodilatation) in response to acetylcholine [7]. The importance of endothelial dysfunction was demonstrated with angiography; during exploration it was seen that a paradoxical vasoconstriction occurred when acetylcholine was administrated in patients with severe atherosclerosis in contrast to the vasodilation in people with normal coronary vessels [2]. This study shows the importance of a deficient response in the circulatory system due to a lack of NO production.

### 3.2. PON1 Function

PON1 is a 43 kDa calcium-dependent glycoprotein formed from 355 amino acids and is a member of the PON family with PON2 and PON3, located on chromosome 7, all with a protein structure. It is mainly produced in the liver, as well as in the kidneys, and is usually transported in plasma, being associated with high-density lipoprotein (HDL). Oxidative stress and free radicals have a major role in many pathologies such as neurodegeneration, liver and cardiovascular diseases, diabetes, and most importantly atherogenesis. All of the above are developed by inflammation (TNF-α, IL-1, and IL-6) and oxidative stress. It is well known that antioxidative substances are needed to prevent these pathologies, and it is believed that paraoxonase enzymes have a protective role due to their ability to contribute to antioxidant and anti-inflammatory pathways, especially paraoxonase 1 (PON1) [9,19,20].

PON1 has calcium ions sites; therefore, it needs calcium for activation [19]. If the calcium is removed, then some of the activities dependent on calcium are inactivated, such as POXase and AREase activity, but the oxidative protection of LDL is still ongoing [21]. The three cysteine parts of its structure are used for its function and for creating a disulfide bridge, but most importantly for their role in protecting against LDL oxidation [22]. It is known that PON1 has a big role in stopping the first stage of foam cell formation, and in atherosclerotic plaque geneses by preventing low-density lipoprotein (LDL) oxidation, macrophage formation, and the production of monocyte chemoattractant protein-1 (MCP-1) [19]. Studies on animals showed a difference between the overexpression of PON1 associated with a higher resistance to inflammation and lipid oxidation, whereas those with lower levels were more susceptible to atherosclerosis [9]. Serum levels are different between people because their levels are modified by diet, environment, and diseases [23]. PON3 has also been studied recently, and it has some of the proprieties of PON1, being involved in the prevention of atherosclerosis, but it only has arylesterase and lactonase activity [20,24]. PON2 is more frequently found in females; it also has oxidative properties, and it might be a reason for the protection against a few cardiovascular and neurological diseases [25]. There are some differences in PON1 function and plasma levels depending on the polymorphism of the genes that code PON1. Some changes can be explained by the polymorphism of the amino acids from position 192 (arginine or glutamine), which in this case resulted in the lowest activity of PON1, or there can be some changes in the 55 positions (leucine or methionine). Coronary heart disease is associated with a higher chance of having the R allele or homozygosity of the L allele [22]. Mohamed et al. demonstrated in the Egyptian population with coronary disease that HDL levels were lower and total cholesterol, triglyceride, and LDL cholesterol were higher in those with the PON1 RR allele than those with PON1 QQ, but there was no correlation with the severity of coronary atherosclerosis [26]. Also, those with the G allele of the rs662 single-nucleotide polymorphism have one more added risk factor to develop coronary artery disease, while rs854560 is not linked with it [27,28].

The PON1 natural structure is based on lactone and lipophilic substrates. The aromatic amino acids from the active sites are responsible for the adherence to lipophilic substrates. Lactonase activity is implicated in the antiatherogenic property of PON1 by reducing the oxidative function of LDL and macrophages [21]. It has been demonstrated that lactonase activity occurs even if it is not connected to the HDL environment [21,29].

PON1 is also a factor that is implicated in the regulation of blood pressure, and the main mechanism is controlled by 5,6-epoxyeicosateroic acid, which might also affect the salt-sensitive, high-pressure mechanism that involves the RAAS pathway and is efficiently targeted in the treatment of high blood pressure [30]. Another study demonstrated the substitutive effect of 5,6-δ-DHTL (5,6-δ-dihydroxyeicosatrienoic lactone) acid and PON1 regarding the vasodilatation mediated by Ca^2+^ in mice [31]. The activity of PON1 is supposed to be present only while it is bonded to the endothelial membrane and, in consequence, endothelial dysfunction leads to the internalization and enzymatic degradation of the enzyme [32].

Paraoxonase 1 is diminished in infectious diseases, it can be used as a marker, and it can lead to lower protection against the microorganism; therefore, it can play a part in the development of antibiotic resistance. In those with severe sepsis, PON1 can be used as a marker serum suggestive of a worse prognosis. Another advantage is that PON1 can fight against endotoxins from Gram-negative bacteria [33,34]

### 3.3. PON1 Implication in Atherogenesis and Cardiovascular Diseases

HDL within normal limits is responsible for cholesterol efflux, for stimulating nitric oxide, prostacyclin (PGI2), and cyclooxygenase with their vessel protective proprieties, and for the anti-inflammatory response via paraoxonase 1 [35]. Also, HDL inhibits the factor that activates plaquettes (PAF); therefore, it plays a role in stopping platelet aggregation [36]. The bonding between HDL and PON1 is influenced by some apolipoproteins such as apoA-1 and ApoJ, which are implicated in metabolic diseases, and the bonding is promoted by scavenger receptor class B type I (SR-BI). Although HDL is necessary for its activity, PON1 is also likely to be found in chylomicrons and VLDL (very low-density lipoprotein), but in smaller concentrations [19]. MPO is a protein that is usually present in healthy people’s plasma in very low concentrations, and levels are elevated in atherosclerotic diseases like acute coronary syndrome or those with metabolic syndrome, and it affects PON1 activity, decreasing its activity [37,38]. A high MPO/PON1 ratio is representative of HDL dysfunction; MPO inactivates the function of HDL and oxidases the tyrosine residue of PON1 that is responsible for the connection with HDL; therefore, it is linked with atherosclerotic disease [37,39]. HDL extracted from those with a higher MPO/PON1 ratio does not have the same anti-inflammatory effects, and in those patients’ serum, the NF-kB pathway is activated [39]. MPO levels can be used in the assessment of the instability of the atherosclerotic plaque according to recent studies [40]. It has been found in mice models that the absence of PON1 leads to lower levels of glycerin, PUFA ratio, 3 hydroxybutyrates, and carnitin, which means lower lipolytic activity and less lipid oxidation [9]. Furthermore, this can reduce the homocysteinilation of proteins with less hcy-thiolactone, and lead to a lower risk for cardiovascular, autoimmune, and neurological disease. It has three main activities: arylesterase (enhanced by ApoA-1), lactonase, and organophosphate activity [41].

Kunutsor et al. demonstrated in a study that PON1 activity is more firmly related to HDL-C and ApoA-1 and has a linear association with CVD (cardiovascular disease). HDL-C is believed to be a stronger risk indicator for CVD than PON1, but another study shows PON1 as a better indicator in men known before with cardiovascular disease [42]. Results from a recent study show that PON1 arylesterase activity in people with coronary arterial disease is lower than in normal people [43].

One of the keys to stopping atherogenesis is to prevent low-density lipoprotein oxidation. Oxidated forms of LDL and HDL cholesterol by ROS are responsible for inflammation, and they interfere with glucosidic and lipid metabolism, especially in the Krebs cycle, glycolysis, and phospholipid metabolism. Therefore, metabolic diseases that lead to high ROS levels decrease PON1 levels and its ability to stop LDL oxidation and the activation of pro-inflammatory cells [9]. Furthermore, low PON1 activity leads to HDL dysfunction, which is responsible for more LDL oxidation and higher LDL levels, since HDL usually has protective antioxidant and anti-inflammatory properties, and its normal function is vital. High oxLDL levels increase MCP-1 and lead to atherosclerosis [9]. PON1 can hydrolyze oxidized fatty acids, triglyceride hydroperoxides, and cholesterylesters, which are implicated in atherosclerosis; therefore, this is one of the mechanisms that saves the LDL from oxidation [44]. Another mechanism is given by PON1’s ability to hydrolyze the macrophage plasma membrane surface of phospholipids when combined with HDL. This process drives n-lysophosphatidylcholine (LPC) production, which inhibits cholesterol fusion [45]. There have been studies on macrophages showing that PON1 reduces macrophage oxidative stress, indicating their role in oxidizing LDL [21]. A study including 3668 patients without acute coronary syndrome who received a coronary angiography had PON and arylesterase activity measured and re-evaluated for 3 years, showing that low levels are associated with a higher chance for major cardiovascular events [46]. Also, it is believed that PON1 activity is already diminished before the acute event, since the activity is lower within 2 h of the symptoms of myocardial infarction. Besides that, PON1 is correlated with the severity of atherosclerosis [47].

PON1 has a beneficial effect and may have implications in reducing diseases by being involved in two significant pathways: peroxisome-proliferator-activated receptor gamma (PPAR-γ) and protein kinase B/nuclear factor kappa–light chain enhancer of activated B cells (AKT/NF-kB). Growth factors and other effectors stimulate all the PON1 pathways, leading to the formation, differentiation, and apoptosis of the cells. IL-1β and TNF-α (tumor necrosis factor) reduce PON1 activity, inhibiting PPAR-α activation via NF-kB and IL-6 (interleukin-6) stimulates PON1 via AKT/NF-kB. NF-kB is a transcription factor and plays a role in the expression of pro-inflammation [48].

During a pro-inflammatory phase with increased IL-6, IL-1 β (interleukin-1 β), TNF-α, serum amyloid A (SAA), and reduced PON1, the mRNA inhibitor of NF-kB translocation to the nucleus or the transient overexpression of IkB partially restored PON1 mRNA levels [49]. PPAR antagonists are widely used drugs in medicine; fibrates as hypolipidemic drugs and thiazolidinediones are used in treating diabetes [50]. The effect of fibrates on PON1 has contradictory results depending on the type, dose, and length of treatment. Fenofibrate increased PON1 activity in people with high levels of cholesterol and ischemic heart disease, while bezafibrate and gemfibrozil for 8 weeks did not have a direct effect on PON1 [49,51]. It is well known that CRP is one of the best markers for inflammation, and Mackness et al. also investigated the PON1: CRP ratio for its clinical relevance to coronary heart disease. The CRP is higher in non-diabetic subjects with coronary heart disease than in people with diabetes, and PON1 is the most reduced, but further investigations are needed to assess if those are the best markers [52].

IL-6, a cytokine known for its implication in atherogenesis, is responsible for increasing the function and protein level of PON1, but this does not apply to IL-1 β or TNF- α in this study. The mechanism behind this process is observed at the transcriptional level by the power of IL-6 to activate the JAK/STAT 3 (Janus kinase) signal pathway and the activation of NF-kB. Therefore, IL-6 helps the binding between NF-kB and the PON1-specific response sequence, and this effect was annulled by inhibiting the IL-6-induced PON1 gene expression in an experimental study with NF-kB inhibitors [41]. On the other hand, another study suggests that IL-6 plays an important role in PON1 activity, showing that the long-term regulation of PON1 by IL-6 was detrimental [53].

Studies on mice showed higher levels of the aortic adhesion molecules P-selectin and ICAM1 mRNA, greater leukocyte adhesion, and higher levels of aortic superoxide, supporting the idea that a lack of PON1 means oxidative stress and the beginning of atherosclerosis. Also, there was less cellular HDL binding. Additional studies with mice models using leptin and LDL (low-density lipoprotein) receptor deficiency, stimulating metabolic syndrome, have led to atherosclerosis, dyslipidemia, and obesity. In this study, an overexpression of PON1 resulted in decreased levels of oxidized LDL and plaque volume [49].

Interestingly, a newer study has found that PON1 and its arylesterase activity were diminished in patients with atrial fibrillation. Still, atrial fibrillation is responsible for the changes in PON1, but not for its activity. The explanation for this finding can be that atrial fibrillation’s risk factors are the same that lead to lower PON1 levels [54].

Reduced PON1 levels were found in patients with carotid stenosis who were symptomatic in comparison to those who did not have any symptoms. Still, the levels were not associated with cholesterol or triglyceride levels, which can be explained because complicated plaques contained more oxidized LDL, and symptomatic patients were older. It is known that PON1 levels are reduced with aging [55]. A recent study on patients with non-ST-segment elevation myocardial infarction showed that diminished PON1 levels were found in those who had an increased risk of death [56].

Recent studies determined that PON genes are not only responsible for protection against atherosclerotic cardiovascular diseases, but also against those that are not atherogenic (Figure 3). They protect against hypertrophy of the heart, a mechanism used for compensation in hypertensive patients but that can lead to heart failure over time [57,58].

### 3.4. PON1 and Diabetes

Type 2 diabetes is associated with increased oxidation with thrombotic results such as hyperaggregability [59]. The effects on glucose metabolism have been studied, finding that PON1 affects insulin sensitivity and glucose tolerance, and that the abnormal PON1 function leads to insulin resistance [19]. The relationship between them is mutual because polymorphism can be a reason for the installation of diabetes and after the onset, PON1 activity will be much lower [60]. The malfunction of PON1 activity has been found in both diabetes mellitus type 1 and type 2. These results are also important for cardiovascular diseases, since diabetes is one of the most important risk factors. Diabetes also has a role in lipidic metabolism, being responsible for increased triglycerides and LDL cholesterol levels and decreased HDL levels. If the triglyceride part of HDL is higher than normal, then the chances of plaque apparition in patients with diabetes are higher [61]. It is well known that HDL plays a role in glucose metabolism by promoting insulin release from pancreatic cells and the conservation of glucose in skeletal muscle [9]. In type 1 diabetes, Vaisar et al. found that high PON1 levels and moderately increased HDL values were responsible for vascular protection without taking into consideration the rest of the lipid metabolism’s markers [62].

Regarding the mechanism, it is believed that increased levels of diacylglycerol caused by high levels of glucose can cause PON1 gene transactivation via the PKC-mediated pathway [45]. The body’s response to insulin by GLUT4 expression is increased by PON1, but PON1 levels are independently reduced by type 2 diabetes, with a decreasing trend as diabetes progresses [63,64]. In diabetic patients, PON1 activity is lower, and it can go even lower in those who are also suffering from microvascular complications of diabetes, leading further to a higher chance of developing atherosclerosis [37]. Studies suggest that low PON1 levels in diabetes are one of the causes of retinopathy and even nephropathy. When insulin resistance occurs, first there are no significant changes in PON1, showing the importance of an early diagnosis and of lifestyle changes and treatment in the first phase [65]. An interesting new study showed that control females had higher PON1 activity than men, but when the comparison between type 2 diabetic patients was made, they had lower levels than men [66].

### 3.5. PON1 and Stroke

Over the years, stroke has become the second cause of death and disability worldwide. More than 15 million people are affected by this disease, defined as an acute syndrome with a focal neurological deficit due to vascular injury located in the central nervous system. The most prevalent type of stroke is ischemic, which comprises approximately 85–90% of all cerebral infarcts. Ischemic strokes are caused mostly by arteriolosclerosis located in the small cerebral vessels; another cause could be a cardioembolism or athero-thromboembolism located in large arteries [67]. Studies of stroke etiology showed an interaction between non-modifiable risk factors (genetic) and modifiable risk factors (environmental factors). Regarding the non-modifiable risk factors, numerous genes are linked to cholesterol metabolism, inflammation, and coagulation. Some genes are involved in the etiology of ischemic stroke. Studies have shown that the genetic variations in PON1 can predict the risk of ischemic stroke and modulate the hydrolyzing effect of PON1 over LDL oxidized phospholipids [68,69]. Although the predictive value of PON1 has not been clearly established yet, compared with healthy people, those who had an acute ischemic stroke had lower HDL levels and diminished PON1 activity, with the 107 T allele being responsible for a higher risk of developing an acute stroke [70]. Also, the rs854560 polymorphism is more prone to ischemic stroke [71]. Carotid atherosclerosis can be a cause of stroke, and PON1 is implicated in intima media thickness, but some studies showed a link between them only in the 55LL genotype. Another study found that the rs662 polymorphism is associated with the severity of carotid stenosis [72,73]. The enzyme PON1 is supposed, in various studies, to be one of the most important markers, along with HDL cholesterol, to limit LDL cholesterol oxidation, preventing first atherosclerosis and second stroke [69,72,74]. Further studies are required to establish a relation between PON1 activity levels and the prognosis of patients with ischemic stroke, and this relationship may have clinical and therapeutic importance in neurological patients [69,74].

### 3.6. PON1 and Other Diseases

Obesity has also been found to be involved in lower PON1 activity and higher levels of HDL and LDL hydroperoxides, meaning more oxidative stress. Also, leptin levels are indirectly proportional to PON1 arylesterase activity, while adipokines are positively correlated with arylesterase PON1 activity [9]. Obesity leads to a higher chance of developing cardiovascular disease and moreover, the body index is directly related to PON1 [65]. One reason can be based on the adipocytes, since they not only store the lipids but also produce pro-inflammatory cytokines and adipokines that inhibit liver PON1 production [75].

Uremic components can interfere with HDL structure, and since in renal disease their elimination is reduced, it will lead to lower PON1 activity. PON3 is also affected by chronic kidney disease. Kotur et al. also studied PON1 levels in COPD patients, who had lower PON1 levels than the control group. Also, the same result has been found in sarcoidosis [76]. A recent review showed that PON1 was lower in the serum of asthmatic patients, but it seemed that it was not correlated with the severity of the disease [77]. The same group investigated its activity in psoriasis with strongly reduced PON1 levels, but no differences were found in arylesterase activity [78]. PCOS can be associated with PON1 deficiency; however, mostly depending on the polymorphism, in Kashmiri women only, 108C/T, but not 55L/M genes, were associated with PCOS, increasing the chance of developing metabolic syndrome and insulin resistance [79]. Nonalcoholic fatty liver disease (NAFLD) is a chronic liver condition associated with excessive fat that builds up in the liver, and it can progress into NASH (nonalcoholic steatohepatitis) and finally to cirrhosis or liver cancer. Kotani et al. proved that PON1 activity is diminished in NAFLD, and they suggested that it can be used as a marker for it [80].

### 3.7. PON1 and Therapy

Statins, inhibitors of hydroxymethylglutaryl-coenzyme A reductase, are the main choices for lowering cholesterol levels, but are also implicated in the stabilization of pre-existent plaques, protection against oxLDL, increased NO effect, and upregulation of eNOS. Also, they block the NF-kB pathway responsible for inflammation [7].

PON1 activity’s response to statin therapy has been studied, with two different outcomes: one study suggested increasing activity, while another study with simvastatin suggested no changes [21]. Deakin et al. studied simvastatin in patients, measuring total cholesterol, LDL, HDL, triglycerides, PON1 concentration, PON1 activity ARE, PON1 activity PONb, specific activity ARE, and specific activity PONb, with higher activity for paraoxon and phenylacetate hydrolysis, both of them being associated with a higher serum concentration of PON1 [81]. Usually, in studies, PON1 activity is measured using paraoxonase activity and arylesterase activity, since AREase is not affected as much as others by genetic polymorphism, so it has a minimal interindividual. Also, it is a reliable marker for measuring PON1 levels and activity [21]. Another study on simvastatin conducted by Kumar showed that LDL levels were diminished with no influence on HDL, but after 4 months of treatment, PON1 levels rose [82]. As far as PON1 polymorphism is considered, studies showed a positive effect on HDL with pravastatin in an R allele carrier than QQ homozygous, and better levels in RR homozygous during simvastatin therapy. DE Souza et al. had a different outcome with RR carriers who did not reach the HDL target as easily as those with the Q allele [83]. Clopidogrel, an antiplatelet therapy, used in the prevention and treatment of patients who undergo coronary angiography with stent implantation, has been a controversial study. Some studies suggest that the PON1 QQ192 genotype might be responsible for some resistance to clopidogrel because this genotype has been associated with stenosis intrastent, but others found no connection [84]. Ticagrelor, used more and more frequently in the past few years in patients with coronary stents, has had auspicious results regarding its effectiveness in protection against another ischemic event, and more importantly, it has the ability to induce PON1, leading to higher serum levels in comparison to clopidogrel [85]. Acetylsalicylic acid, a highly used drug in cardiovascular medicine, known for its protective antithrombotic and anti-inflammatory effects, had a positive impact on PON1 in studies on people with an atherosclerotic coronary [86]. Ezetimibe, a drug used in combination with statin therapy in patients whose target cholesterol level cannot obtained with statins alone, is used for stopping cholesterol absorption and was studied in 2010, showing that antioxidant parameters such as PON1 and TAC levels were increased substantially [87].

### 3.8. PON1 and Lifestyle

Diet plays a crucial role, since increased lipid intake leads to inflammation and higher levels of ROS (reactive oxygen species), pro-inflammatory cytokines, IL-1, and IL-6, all of which result in lower PON1 liver secretion and decreased PPAR-γ. On the other hand, a high-sucrose diet has had controversial results. In vivo studies demonstrated that high sucrose intake is responsible for hyperlipemia and increased oxidative stress, also suggesting lower PON1 activity [9]. Another study has found that a high-sucrose diet leads to high PON1 activity after 3–5 weeks [88]. The Mediterranean diet, recommended for the prevention of cardiovascular diseases, has resulted in higher postprandial PON1 activity, which might be the result of using extra virgin olive oil [9]. Studies on mice on an atherogenic diet demonstrated no changes in gene expression, but lower activity of PON1 [49]. Meals cooked in deep-fried oil lowered HDL and PON1 4 h after their consumption, and the effects lasted approximately 8 h [49]. Extra virgin olive oil is recommended to be used instead of sunflower oil because studies have shown numerous beneficial effects like reducing inflammation (also CRP and IL-6), oxidative stress, and increasing the levels and function of HDL [22]. As expected, since cigarette smoking increases oxidative stress, in smokers there was also lower PON1 activity [89,90]. As far as alcohol consumption is concerned, the results are interesting. A study showed that chronic light alcohol intake raises PON1, while heavy drinking leads to lower PON1 expression. This explication comes from the fact that ethanol, by stimulating PKC, will raise PON1, but heavy consumption leads to PKC overexpression, which will inhibit PON1 [45,91]. Also, red wine, which contains antioxidant components, has led to higher HDL and lower LDL levels, mainly due the effect of the alcohol, as described, and the effect of resveratrol on PPAR-γ receptors [45,49]. PON1 is also modulated by physical activity, with lower levels in people who are sedentary and obese. Still, it has also been found that 192QQ subjects had higher levels compared to R carriers, meaning it is also dependent on some genes [49,92,93]. The main functions of PON1 are included in Table 1. Aging is associated with a decline in cell function, with a higher susceptibility to increased oxidative stress and inflammation; therefore, lower levels of PON1 are expected as well [15].

## 4. Plant Extracts and Phytochemical Compounds with a Positive Effect on PON1

The effects of some plants regarding PON1 have been studied, and the results are encouraging for more studies to be conducted. Some plant extracts are rich in vitamins, carotenoids, and polyphenols. Polyphenols are divided into flavonoids and non-flavonoids, and they contain a minimum of one aromatic ring. Polyphenolic compounds extracted from botanical sources are known for their antioxidant proprieties, but it is necessary to elucidate the mechanisms based on which they can be explained. Thus, we start from the premise that hydroxyl groups serve as efficient hydrogen donors and can engage with reactive oxygen and nitrogen species. This interaction is followed by a termination reaction that effectively stops the generation of free radicals. As a result, the initial reactive species are transformed into an antioxidant radical with significantly improved chemical stability. Due to their structural characteristics, phenolic compounds are available for interaction with different protein structures, through hydrophobic benzene rings and hydrogen bonds, as well as the potential of phenolic hydroxyl groups. As a result, these phenolic substances can be characterized by the ability to act as antioxidants by inhibiting specific enzymes involved in the generation of radicals, such as different cytochrome P450 isoforms, lipoxygenases, cyclooxygenase, and xanthine oxidase. One of the mechanisms of action is the upregulation of a ligand-activated transcription factor via AhR that can associate with the xenobiotic responsive elements or ARNT and bind to polyphenols or other substances in the presence of PON1. Studies on AhR found that it can be implicated in protection from cardiovascular disease, but depending on the ligand, it can also be implicated in atherosclerosis, so the results are controversial because studies showed the induction of PON1, but at the same time they can inhibit PON1, since they can be responsible for oxidative stress. AhR promotes atherosclerosis in three ways: when combined with NF-kB, it will lead to monocyte chemotaxis; AhR stimulates macrophages to absorb oxLDL to create foam cells, increasing the proliferation of vascular muscle cells and will activate NADPH oxidase which will finally lead to the production of ROS that will destroy the endothelial cells. Also, the transcription factor sterol regulatory element-binding protein 2 (SREBP-2), implicated in cholesterol synthesis, will bind with the PON1 promoter, linked with the MAPK cascade, and finally activate the PKC (Figure 4). JNK can also upregulate PON1 by stopping the production of ROS [44,45,94,95,96]. Most polyphenols express the anti-inflammatory and antiatherogenic effect by increasing HDL, and the functional HDL is the one bound with PON1. They will stop the oxidation of LDL, increase the production of NO, prevent the production of pro-inflammatory substances, lower the intensity of the oxidative stress that affects the macrophages, inhibit the circulating lipopolysaccharides (LPS), and inhibit the signaling of T and B cells, known for their implications in inflammation [19,97,98]. AMPK activation will stimulate LPL, which will help with the metabolization of triglycerides, thereby lowering its levels, as well as suppressing PPAR-γ, blocking the formation of adipocytes and lipogenesis [99].

Nuts were evaluated, and a study on pistachio had a positive outcome on rats, with increased HDL and PON1 levels [100]. The mechanism behind it involves stopping the degradation of the nuclear kappa inhibitor, TNF-α, and interleukin-1 beta (IL-1β) [101]. Avocado had similar effects but lowered triglycerides, while acai had the same effects as pistachio [49,102,103]. Pistachios have also been studied by showing they can be responsible for reducing arterial stiffness [104]. Auspicious results were seen in experiments on rats with *Aronia melanocarpa* Michx.-Elliott., a phenolic substance known for its antioxidant and atherosclerotic activity, and pomegranate juice, both leading to higher PON1 levels. Aronia indirectly decreases the activity of pro-inflammatory cytokines and pro-oxidative enzymes like IL-1β, TNF-α, and xanthine, and increases the activity of anti-inflammatory cytokines and antioxidant enzymes like IL-4 and superoxide dismutase. Pomegranate juice, rich in tannin, anthocyanin, quercetin, and resveratrol, acts to prevent inflammation by inhibiting MDA and stimulating CAT, MPO, superoxide, and glutathione peroxidase and by stimulating PPAR-γ inactivation and PKA-cAM, which upregulates PON1 [45,105,106,107,108]. Resveratrol has a positive effect on PON1 and is also known for its capacity to inhibit lipid peroxidation and platelet aggregation. It also interacts with the NF-kB pathway. Some experiments on animals showed that LDL was diminished and HDL and PON1 activity were raised accompanied by reduced glycemic levels. Also, it inhibits mTORC1, which is responsible for foam cell formation, and diminishes the macrophage percent from plaques [19,95,109]. Curcumin, used as a spice, has been proven to be an excellent anti-inflammatory and antioxidant substance, as well as an antiaging element, especially in diseases that are more frequent in adults like cancer and cardiovascular and neurodegenerative diseases. The pathway responsible for its effects implicates the AhR receptor, which will act on CYP1A1 [110]. *Securigera securidaca*’s alcohol extract was used in some studies. One of the studies suggested that oral administration in rat models does not affect PON1 [111], but a more recent study suggested that higher doses can lead to normal PON1 levels after 35 days of use [112]. *Moringa oleifera* L. protects the liver mitochondria against oxidative stress and promotes vasorelaxation by suppressing calcium channels and delaying the onset of diabetes. The oral dosage used on mice resulted in increased activity of PON1 and CAT (chloramphenicol acetyltransferase), and its component beta-sitosterol was responsible for an increase in HDL levels and decreased triglyceride and LDL serum levels [9,113,114]. Luteolin (3′,4′,5,7-tetrahydroxyflavone) is another beneficial flavonoid, which has been used for centuries, is found in green pepper and chamomile tea, and is known for its antidiabetic, anti-inflammatory, and antitumor proprieties. The anti-inflammatory property is exhibited by suppressing the TNF-α-induced expression of MCP-1, VCAM-1, and the IKBα/NF-kB pathway. Also, another lutein form can inhibit the JAK/STAT3 pathway, thereby inhibiting IL-1β [115,116]. Extra virgin olive oil is composed not only of MUFA but also has in its composition more than 30 polyphenols, and the most active is hydroxytyrosol (HT). HT, once arriving in the intestine, will be metabolized, hydrolyzed, and conjugated. Most frequently, it will be in its conjugated glucuronide forms. HT inhibits the production of reactive oxygen species and activates forkhead transcription factor (FOXO) via AMPK, and FOXO3a can directly stimulate antioxidant enzymes, protecting from oxidative stress. Moreover, activating the PI3K/protein kinase B (AKT) pathway and extracellular signal-regulated protein kinases ½ (ERK1/2) will stimulate the action of nuclear factor erythroid 2 (Nrf2), with the final result being the stimulation of heme oxygenase-1 (HO-1), known for its property to repair endothelial damage and to inhibit oxidative stress. Regarding the anti-inflammatory action of HT, it’s known that the mechanism is based on the inhibition of NF-kB, I-kBα, and STAT-1α. Polyphenols from olive oil were also responsible for diminished LDL particles and it was observed that they are preventing the formation of foam cells. As explained above, if the endothelium is not dysfunctional, the inflammation is not present, and the lipid levels are in normal range atherogenesis is not likely to occur. A study demonstrated that a diet rich in high-polyphenol olive oil for 3 weeks stimulated PON1 and PON3 activity [117]. Another medicinal plant of interest is *Rumex acetosa* L., also known as sorel, which is a perennial plant rich in polyphenols, known for anti-inflammation, antioxidant, antibacterial, antitumoral, antidiarrheal, anthelminthic, and cardiovascular protective roles, and has been used as an herbal remedy for a long time [118]. Alzoreky and Nakahara (2001) investigated the antioxidative activity of *Rumex acetosa* L., finding a powerful radical activity. Resveratrol is known to possess anti-inflammatory, antioxidative, and antiplatelet proprieties by increasing PON1 activity and by antagonizing gene transcription such as activator protein-1 or NF-kB [119]. *Rumex acetosa* not only reduced LDL cholesterol, total cholesterol, and liver enzyme levels and increased HDL levels, but also improved glucose tolerance and helped the storage of glycogen in the liver. Also, studies on rats suggested that injecting Rumex methanol extract has a vasorelaxant activity by reducing arterial hypertension [120]. One of the most important properties is the inhibition of collagen-stimulated platelet aggregation in a dose-dependent manner by *Rumex acetosa* extract [121]. Another important property is its antihypertensive effect with a dose-dependent response. This can be endothelium-dependent action mediated by NO, L-NAME (nitro arginine methyl ester), indomethacin, and atropine. As far as the independent effects are concerned, it has been found that they are based on calcium inhibition, an effect similar to a calcium channel blocker [122]. Also, a study included three plants, one of them being *Rumex acetosa*, suggesting an antiproliferative effect provided by resveratrol, vanillic acid, sinapic acid, and catechin [123]. The methylene chloride fraction of *Rumex acetosa*, but mostly emodine, is believed to have an antimutagenic activity and cytotoxicity against a few types of cancer cells [124]. Positive action in terms of PON1 activity was also found in extracts of *Aronia melanocarpa* Michx.-Elliot., rich in flavonoids, with genistein being a promising bioactive agent. Flavonoids are found in the composition of all active extracts, which is beneficial in the long-term prevention of chronic conditions [105].

Of interest to the researchers was the exploration of the effect of onion extract, namely quercetin and catechin, in regulating PON1 expression, and its correlation with oxidized LDL levels. This study was performed in male Wistar rats in which oxidative stress was induced using mercuric chloride (HgCl_2_). The treatment lasted four weeks. As a result, it was observed that free-radical-scavenging activity as well as PON1 activity decreased, along with the increase in susceptibility of LDL to oxidation, but there was also an increase in the plasma levels of malondialdehyde. The administration of onion extract significantly attenuated the unwanted effects of HgCl_2_. This occurred by upregulating the activity of PON1 and the radical-scavenging activity. In this way, LDL oxidation and lipid peroxidation were protected. These effects were similar in the group treated with quercetin, and to a lesser extent in the group treated with catechin [125].

Also of interest is the in vitro study of some plant extracts on human carbonic anhydrase isoforms (hCA I and hCAII) and the activity on paraoxonase 1 (PON1). The research harmonizes the results of determining the activity of selected plant extracts: *Alcea rosea* L., *Foeniculum vulgare* Mill., *Elettaria cardamomum* L., *Laurus azorica* L., and *Lavandula stoechas* L. Extracts were made with methanol, ethanol, and water, and finally used to determine the concentration-dependent degrees of inhibition for hCA I and hCA II isozymes and PON1. Thus, IC50 values (mg/mL) were obtained for each extract. The methanolic extract of *Elettaria cardamomum* had the highest inhibitory effects (0.032 mg/mL) compared to the aqueous extract of *Alcea rosea* (1.721 mg/mL). Another observation is that the aqueous extracts of the plants showed a lower inhibitory impact compared to the extracts in methanol and ethanol [126]. A series of data about the involvement of bioactive compounds in supporting the protective role of paraoxonase 1 is included in Table 2.

## 5. Conclusions

Endothelial dysfunction is the preclinical stage in the development of atherosclerosis in correlation with low NO levels. PON1 levels and activity are downregulated by oxidative stress and inflammation. Mediterranean diet, low body mass index, and a physically active status can lead to higher PON1 concentration and therefore a higher anti-inflammatory and protective effect, but it is important to keep in mind that PON1 is also influenced by some genes found in polymorphism studies. One of the biggest roles of PON1 is the protection against atherosclerosis, which is given by its ability to bond with HDL and reduce LDL and triglycerides serum levels, thereby fighting against cardiovascular disease while reducing oxidative stress. More interest should be paid to determining PON1 levels, as it is one of the best markers to be taken into consideration when assessing inflammation and the severity of atherosclerosis. The focus should also be on introducing into our diet the plant extracts that have been found to enhance PON1 levels, whose beneficial activity has been proven, and discovering novel substances that can modulate PON1, thereby delaying the onset of diabetes and cardiovascular diseases. In conclusion, dietary polyphenols such as pomegranate juice, nuts, resveratrol, *Aronia melanocarpa* Michx.-Elliott, *Rumex acetosa* L., luteolin, and many others have been proven to modulate the induction of PON-1 using different mechanisms of action, but it is strongly suggested that they lead to the same result, a reduced state of pro-inflammation (lower IL-1β, TNF-α IL-6) and lower chance atherosclerosis. Furthermore, novel studies should focus on assessing the metabolism, bioavailability, adverse effects, duration, and the mechanism that PON1 is using in preventing diseases.

## Figures and Tables

**Figure 1 ijms-25-02962-f001:**
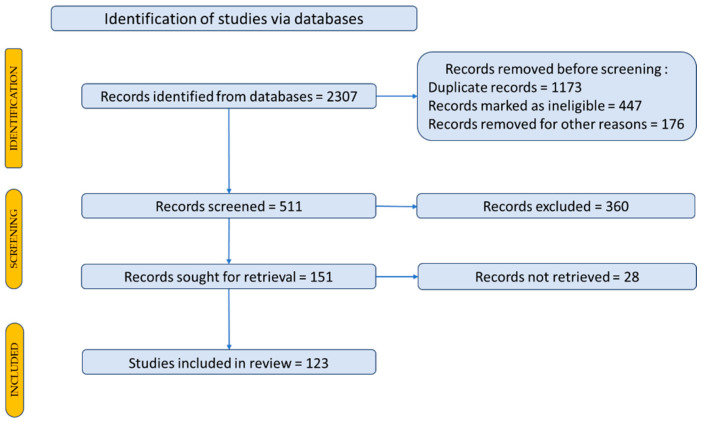
Prisma flow diagram for description of the selection process of the bibliographic sources.

**Figure 2 ijms-25-02962-f002:**
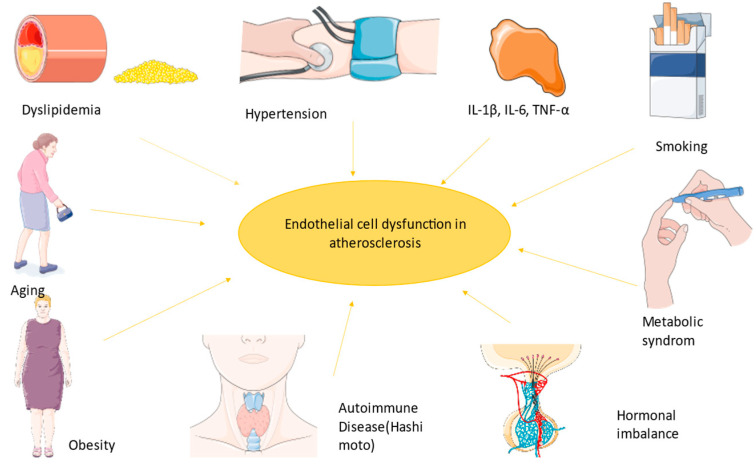
The most important factors implicated in endothelial dysfunction led to a state prone to atherosclerosis. The figure was composed using Servier Medical Art templates and PowerPoint, Servier Medical Art by Servier is licensed under a Creative Commons Attribution 3.0 Unported License (https://creativecommons.org/licenses/by/3.0/ Accessed on 23 February 2024).

**Figure 3 ijms-25-02962-f003:**
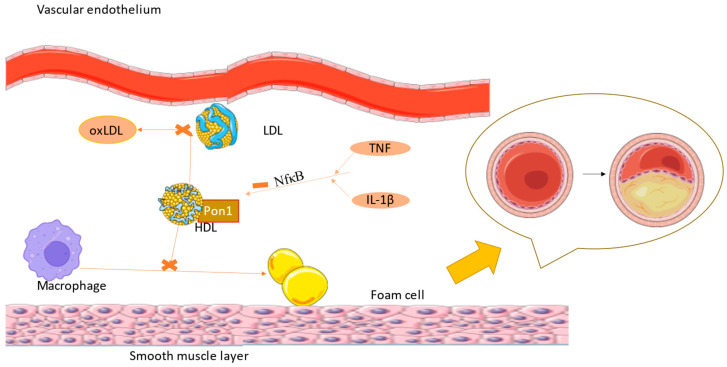
IL-1 β and TNF-α inhibit PON1 function via the NF-kB pathway, leading to higher levels of oxLDL, with atherosclerosis being the consequence (the figure was composed using Servier Medical Art templates and PowerPoint, Servier Medical Art by Servier is licensed under a Creative Commons Attribution 3.0 Unported License (https://creativecommons.org/licenses/by/3.0/) Accessed on 23 February 2024).

**Figure 4 ijms-25-02962-f004:**
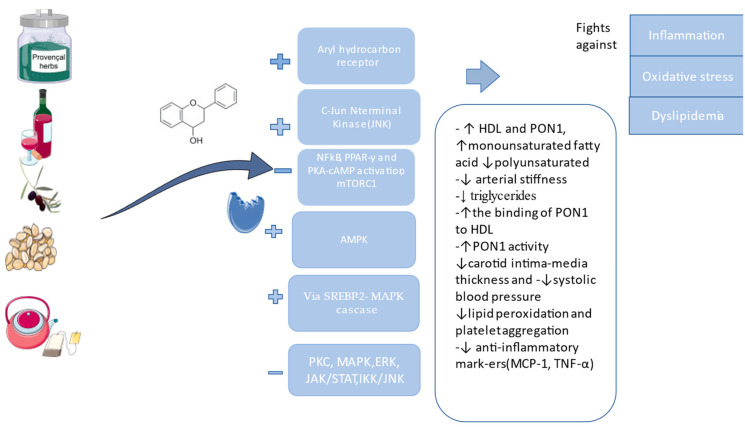
The pathways of polyphenols regarding the modulation of PON1 (the figure was composed using Servier Medical Art templates and PowerPoint, Servier Medical Art by Servier is licensed under a Creative Commons Attribution 3.0 Unported License (https://creativecommons.org/licenses/by/3.0/ Accessed on 23 February 2024).

**Table 1 ijms-25-02962-t001:** PON1 main functions.

1. Preservation of HDL by fighting against its oxidation [23] -higher cholesterol efflux [23]
2. Preventing LDL oxidation [23] -less oxidized lipids, which are responsible for inflammation [49] -lower LDL levels [23]
3. Protects against insulin resistance [9]
4. Ameliorates effects of oxidized LDL [23] -↓ inflammatory and cytotoxic oxidized phospholipids [23] -↓ LDL uptake by macrophages [23] -↓ monocyte transmigration induced by oxidized LDL [23]
5. Atheroprotective [21] -decreases lipid peroxides in atherosclerotic lesions [23] -reduces macrophage oxidative stress and the ability of macrophages to oxidize LDL [21] -contributes to the metabolism of homocysteine thiolactones [21] -prevents the oxidative inactivation of lecithin cholesterol acyltransferase [21] -reduces monocyte–macrophage inflammatory response [21] -reduces foam cell formation [72] -restores normal endothelial function [72]
6. Antiapoptotic [76]
7. Vasodilative [76]

**Table 2 ijms-25-02962-t002:** Plant extracts are involved in supporting the protective role of paraoxonase 1.

Plant Extracts	Compounds	Mechanism of Action	Refs.
*Pistachia vera* L.	lutein, β-carotene, and γ-tocopherol in addition to containing selenium, flavonoids, and proanthocyanidins	-increases HDL and PON1, enhancing antioxidative defense by ↑ monounsaturated fatty acid and ↓polyunsaturated and saturated fatty acid uptake -reduces arterial stiffness -degrades NF-kB, TNF-α, and IL-1β	[100,101,104]
*Euterpe oleracea* Mart.	polyphenolic flavonoid–anthocyanins (cyanidin 3-rutinoside, cyanidin 3-glucoside, cyanidin 3-sambubioside, peonidin 3-rutinoside, peonidin 3-glucoside) and flavonoids (chrysoeriol, luteolin orientin, homoorientin, quercetin, isovitexin, vitexin, dihydrokaempferol)	-long-lasting vasodilation dependent on activation of the nitric oxide–cGMP pathway using endothelium-derived hyperpolarizing factor (EDHF)	[102]
*Persea americana* Mill.		-↓triglycerides -↑HDL-cholesterol and activity of PON1	[49]
*Aronia melanocarpa* Michx.-Elliott	anthocyanins (cyanidin–glucoside, cyanidin–arabinoside), phenolic acids, proanthocyanidins, and flavonols (quercetin–rutinoside (rutin quercetin)) -tannins, anthocyanins (punicalagin, punicalain, gallic acid, and urolithins A and B)	-↑binding of PON1 to HDL -↑PON1 activity -↓carotid intima media thickness -↓systolic blood pressure -↓lipid peroxidation and platelet aggregation -modulates NF-kB pathway	[105,106,107]
*Securigera securidaca* L.	phenolic acids, cardenolides, flavonoids, pentacyclic triterpenoid-type saponins, steroidal	-↓oxidative stress -↑antioxidant defense system -regulates the balance between antioxidant/oxidant -antihyperlipidemic -↑PON1 -↓markers of lipid peroxidation (MDA) -no effect on hs-CRP, TNF-α	[19,111]
*Moringa oleifera* Lam.	phenolic acids, flavonoids, tannins, triterpenes, glucosinolates	-protects against oxidative stress in the liver cells -vasorelaxation inhibits Ca channels -reduces diabetes onset -↑PON1, CAT, and HDL, -↓triglyceride and LDL	[99]
*Rumex acetosa* L.	Phenols (cis-resveratrol, trans-resveratrol, vanillic acid, sinapic acid, catechin rutin, hyproside, quercetin, avicularin, orientin, and iso-orientin) anthraquinones, tannins, alkaloids, stilbenes, lignans, naphthalenes, diterpene terpenes	-↓LDL cholesterol, -↓total cholesterol, -↓liver enzyme levels -↑HDL levels but improved glucose tolerance -↑storage of glycogen in the liver -antimutagenic activity -inhibit thrombus formation (inhibits the phosphorylation of ERK1/2 and JNK, lowered ATP release in collagen-stimulated platelets)	[119,120,121]
*Allium cepa* L.	quercetin and catechin	-protection of LDL oxidation and lipid peroxidation	[125]
*Olea europaea* L.	mono- and polyunsaturated fatty acid	-↑PON1 activity (PON and arylesterase)	[49]
Resveratrol(3,5,4′-trihydroxystilbene)		-modulates NF-kB pathway -↓LDL, ↑HDL and PON1 activity -↓anti-inflammatory markers (MCP-1, TNF-α)	[19]
Luteolin(3′,4′,5,7-tetrahydroxyflavone)		-suppresses TNF-α, MCP-1, VCAM-1, IKBα/NF-kB pathway -inhibits the JAK/STAT3 pathway, IL-1β	[115,116]

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
