# Peer review of "The Role of Polyphenols in Modulating PON1 Activity Regarding Endothelial Dysfunction and Atherosclerosis"

_ijms, 2024, doi:10.3390/ijms25052962_

Round 1
Reviewer 1 Report
Comments and Suggestions for Authors
Dear Authors,
Paraoxanase enzyme is associated with HDL and provides antioxidant properties to HDL. This enzyme also displays arylesterase and lactonase activities together. This enzyme protects LDL from oxidation by hydrolyzing specific lipid peroxides in oxidized LDL and has antiatherogenic properties. With this feature, it has protective properties against many diseases such as dyslipidemia and atherosclerosis, especially cardiovascular diseases. The authors tried to present the basic features of the PON1 enzyme and its mechanism in the manuscript. In particular, they summarized the principle effects of PON1 pathways. However, compared to other review articles, they did not present any innovations, a new pathway or molecular mechanism that could be effective in this field. Manuscript should start with listing the main objectives of the review study, followed by comment how the draft supports the results in the new literature. This manuscript has significant limitations, has many shortcomings and is in need of substantial improvement.
Also, the manuscript is very poorly written - the text is overly repetitive; there are extensive grammatical issues and a general weakness in the written English.
The quality of images (figures) should be improved.
The figures in the manuscript are not compatible with the way described in the manuscript. The mentioned mechanisms are not expressed in the figures. The figures representations do not show the intended mechanisms and are considered in a simplistic manner.
Additionally, the literature is not up to date. At least 30% of the references must be current.
Comments on the Quality of English Language
the manuscript is very poorly written - the text is overly repetitive; there are extensive grammatical issues and a general weakness in the written English.
Author Response
Thank you for the appreciations.
We are sending you our answers attached.

Reviewer 2 Report
Comments and Suggestions for Authors
In my opinion, the article submitted for my review requires much work to be suitable for publication.
Articles of this type - a review article- should be a kind of compendium, pointing out and summarizing the latest discoveries and different opinions and studies. This article is not quite a good example of that.
The most exciting part is chapter 4.
The remaining part seems relatively weak; it does not provide summaries and mechanisms but often presents obvious and commonly known aspects, like endothelial dysfunction - described generally.
The goal is unclear to me: "This review summarizes the principles effects of PON 1 ways to modulate PON1 expression and some potential treatments"
I didn't find an answer to this goal in the article. Besides, it is not formulated correctly.
Figure 1 is straightforward, and honestly, I don't know what it brings and how polyphenols - by what mechanism - affect PONs.
Nothing follows from this drawing.
Figure 3 is of poor quality and contains super obvious aspects; there is no connection with the topic, i.e., PON-1. This is the topic - atherosclerosis.
Generally, the article has too many side threads, such as endothelial dysfunction. Maybe the authors should focus on Plant Extracts and Phytochemical Compounds with a Positive Effect on the PON1 and expand it.
In my opinion, the article is not suitable for publication in its current form in such a good journal. It is weak in cognitive and technical terms and contains numerous linguistic errors. In the article, only a few aspects are relevant to PON-1, but they are covered too generally.
Moderate editing of the English language is required.
Author Response

(The authors gave the same response as above.)

Round 2
Reviewer 1 Report
Comments and Suggestions for Authors
I would like to thank the authors for their efforts, but the title, purpose of the article and what is presented in this direction are not compatible with each other.
First of all, 1) giving a summary of the current state of knowledge regarding paraoxanase 2) its biological potential and antiatherogenic effect (explain with figures) 3) polymorphisms of paraoxanase and paraoxonase in cardiovascular diseases 4) outlining the PON1 activity modulators and epigenetic factors 5) describing the protective and preventive roles of HDL and PON1 in the accumulation of lipid peroxide (mechanisms can be presented with figures) 6) Most importantly, in keeping with the main goal, mechanisms explaining how plant extracts support PON1's involvement in protecting atherosclerosis through its anti-atherogenic properties must be provided. I would like to emphasize once again that the importance of choosing these aspects should be emphasized and the purpose should be listed. This manuscript has not examined the above-mentioned properties of PON1 and the molecular mechanism by which it acts. Providing only clinical results and superficial results in the text and tables are not sufficient for this review article. The authors of the selected articles should expand the scope of the review in order to better understand the subject, and more in-depth examinations should be made and presented in the manuscript. Specifically, the purpose and hypothesis should be added, the title should be modified, and the presentation should be made in that sequence. The findings analyzed in plants with PON1 expression modulation and its possible targets need to be emphasized in full, along with the underlying mechanisms. Otherwise, I would not recommend this article to be published in this journal.
Comments on the Quality of English LanguageMinor editing of English language required
Author Response
Esteemed reviewer,
We the authors greatly appreciate your recommendations, therefore we send you our answers in the hope that we have lived up to your expectations.
First of all,
1) giving a summary of the current state of knowledge regarding paraoxanase
2) its biological potential and antiatherogenic effect (explain with figures)
3) polymorphisms of paraoxanase and paraoxonase in cardiovascular diseases
4) outlining the PON1 activity modulators and epigenetic factors
5) describing the protective and preventive roles of HDL and PON1 in the accumulation of lipid peroxide (mechanisms can be presented with figures)
6) Most importantly, in keeping with the main goal, mechanisms explaining how plant extracts support PON1's involvement in protecting atherosclerosis through its anti-atherogenic properties must be provided. I would like to emphasize once again that the importance of choosing these aspects should be emphasized and the purpose should be listed.
Response: Thank you for your suggestion, we, the authors, tried to respect the order you suggested with small changes like starting with endothelial dysfunction for a better understanding of the effects of PON1.
This manuscript has not examined the above-mentioned properties of PON1 and the molecular mechanism by which it acts. Providing only clinical results and superficial results in the text and tables are not sufficient for this review article. The authors of the selected articles should expand the scope of the review in order to better understand the subject, and more in-depth examinations should be made and presented in the manuscript. Specifically, the purpose and hypothesis should be added, the title should be modified, and the presentation should be made in that sequence. The findings analyzed in plants with PON1 expression modulation and its possible targets need to be emphasized in full, along with the underlying mechanisms. Otherwise, I would not recommend this article to be published in this journal.
Response: Thank you for your recommendation, you were right, we added the principle mechanism of polyphenols regarding the modulation of paraoxonase and listed the purpose and hypothesis. As far as the title is concerned we changed it , and it suits better the article (The role of polyphenols in modulating PON 1 activity regarding endothelial dysfunction and atherosclerosis)
Reviewer 2 Report
Comments and Suggestions for Authors
Dear Authors,
Thank you for the changes, but the article is still not very in-depth, which it should be - since it is supposed to be an overview of what is known so far. Let me emphasize once again that perhaps one aspect should be selected. Title: An overview of Paraoxonase 1 in cardiovascular disease - does not reflect the content of the article and does not meet the purpose. This should be considered here. The figures have been improved graphically, but they provide similar content. In my opinion, this is not a sufficient improvement to allow the article to be published. I don't understand the response to my comment: The remaining part seems relatively weak; it does not provide summaries and mechanisms but often presents obvious and commonly known aspects, like endothelial dysfunction - described generally. Response: Thank you for the amendment; the articles we reviewed did not study the molecular mechanism that led to the increased PON1 levels; instead, they focused on the clinical results, but we introduced now some mechanisms in the text and also in the tables. Perhaps the scope of this review should be expanded, rather than replying that the authors of selected articles did not research this. This greatly narrows the scope of this article. This is a review article sent to a very good magazine that has a demanding readership. The current article definitely needs to be more in-depth. The title should either be reformulated or the scope of keywords should be expanded, mechanisms, hypotheses, and different opinions should be added - otherwise I would not recommend publishing this article in such a good journal.
Comments on the Quality of English LanguageMinor editing of English language is still required.
Author Response
Esteemed reviewer,
We the authors greatly appreciate your recommendations, therefore we send you our answers in the hope that we have lived up to your expectations.
Reviewer: Thank you for the changes, but the article is still not very in-depth, which it should be - since it is supposed to be an overview of what is known so far. Let me emphasize once again that perhaps one aspect should be selected. Title: An overview of Paraoxonase 1 in cardiovascular disease - does not reflect the content of the article and does not meet the purpose. This should be considered here. The figures have been improved graphically, but they provide similar content. In my opinion, this is not a sufficient improvement to allow the article to be published. I don't understand the response to my comment: The remaining part seems relatively weak; it does not provide summaries and mechanisms but often presents obvious and commonly known aspects, like endothelial dysfunction - described generally. Thank you for the amendment; the articles we reviewed did not study the molecular mechanism that led to the increased PON1 levels; instead, they focused on the clinical results, but we introduced now some mechanisms in the text and also in the tables. Perhaps the scope of this review should be expanded, rather than replying that the authors of selected articles did not research this. This greatly narrows the scope of this article. This is a review article sent to a very good magazine that has a demanding readership. The current article definitely needs to be more in-depth. The title should either be reformulated or the scope of keywords should be expanded, mechanisms, hypotheses, and different opinions should be added - otherwise I would not recommend publishing this article in such a good journal.
Response: Thank you for your answer and for your recommendation, we changed the title, and it suits better the article (The role of polyphenols in modulating PON 1 activity regarding endothelial dysfunction and atherosclerosis). Also, we introduced the principle mechanism, as you suggested, of polyphenols regarding the modulation of PON1. As far as different opinions are regarded we have in the article different studies including AhR receptor-pathway where some studies found it to be anti-atherogenetic while others found it to be prone to atherosclerosis, depending on the ligand and the same regarding statins where different outcomes were found in studies. Lastly, we introduced the hypothesis and reformulate the aim of the article.
Round 3
Reviewer 1 Report
Comments and Suggestions for Authors
Dear Authors,
Thank you for the edits and changes. The shortcomings noted in the manuscript, however, were not entirely eradicable. Apparently, the authors also stated that they tried to comply with the suggested order with minor changes to eliminate the deficiencies. I appreciate the considerable efforts of the authors. However, I would like them to criticize the manuscript they submitted to this journal of such good quality in more detail and completely.
Most importantly, they should elucidate the mechanism by which plant extracts prevent the occurrence of atherosclerosis through the anti-atherogenic properties of PON1 and the molecular mechanism that causes increased PON1 levels. They should especially show this on the figure. They must also improve each image they present.
I kindly request you to prepare your manuscript taking into account these and previous suggestions and submit your answers accordingly. Otherwise, I would not recommend this article to be published in this journal.
Additionally, the following bold sentence regarding Figure 2 in the material method section should be checked again. This sentence refers to "Figure 1" instead of "Figure 2".
“In Figure 2 we have presented a distribution for the items included in our documentation.”
Author Response
We the authors greatly appreciate your recommendations, therefore we send you our answers in the hope that we have lived up to your expectations.

Reviewer 2 Report
Comments and Suggestions for Authors
The article has been corrected according to my suggestions.
I have a few more minor comments.
Fig. 4 is illegible in places.
Additionally, I suggest authors adding a list of abbreviations.
Author Response

(The authors gave the same response as above.)
